# A Narcissism/Vanity Distinction? Reassessing Vanity Using a Modern Model of Narcissism Based on Pride, Empathy, and Social Behaviors

**DOI:** 10.3390/bs13090762

**Published:** 2023-09-14

**Authors:** Madison A. Wheeler, Lawrence R. Burns, Paul Stephenson

**Affiliations:** 1Psychology Department, Grand Valley State University, Allendale, MI 49417, USA; wheelmad@mail.gvsu.edu; 2Statistics Department, Grand Valley State University, Allendale, MI 49417, USA; stephenp@gvsu.edu

**Keywords:** grandiose narcissism, vulnerable narcissism, narcissism spectrum model, appearance vanity, achievement vanity, pride, empathy

## Abstract

The primary purpose of this study is to evaluate the role of vanity in its longstanding theoretical association with narcissism. This is particularly germane, as the conceptualization and measurement of narcissism have evolved in recent years. This is observed in the development of spectrum and/or dimensional models of narcissism, concomitant with the conceptual developments of vanity that have emerged since its original inclusion in the Narcissism Personality Inventory. Specifically, our research question evaluated whether vanity remains as traditionally construed, i.e., as a facet of narcissism, or is better conceptualized as a distinct construct separated from the earlier models of narcissism and therefore provide novel implications in understanding personality and social behavior. Based on the traits of pride, empathy, and several social behavior variables, it was hypothesized that a differentiation between narcissism and vanity would be observed. The participants were 441 undergraduate students from a large public midwestern university who participated in a self-report online survey. Correlation and regression analyses were conducted. The results revealed that the characterization of vanity is limited to pride and an absence of empathy and social behavior. Findings reaffirm behavior patterns of grandiose and vulnerable narcissism regarding selflessness, image management, and sensitivity to others. The core motivations of vanity are explicated as based on social comparison theory to assess one’s subjective and/or objective value though it is not characterized as a “social” trait or by social behavior, in contrast to how narcissism is characterized. Based on our findings and an improved understanding of the narcissism model, we conclude that vanity is more closely related to the grandiose dimension of narcissism and discuss how the underlying motivations of vanity improve our understanding of grandiose narcissism. We discuss the implications that these findings provide to the developing, modern conceptualizations of narcissism and affirm and expand our understanding of vanity in personality.

## 1. Introduction

### 1.1. Narcissism

The word “narcissism” originates from the Greek myth of Narcissus. In the story, Narcissus rejects Echo, the nymph, and the gods punish him by causing him to fall in love with his reflection [1]. Thus, the depiction of the self-absorption and egotism of Narcissus prompted the term “narcissism” that we now use to assess and describe personality composition or pathology. Narcissism is a multi-faceted and relatively stable personality construct traditionally characterized by self-entitlement, self-importance, and excessive preoccupation with one’s own needs at the expense of others [2,3,4]. While narcissism is commonly perceived as a pathology, it can be found in healthy individuals at much lower levels than in individuals with narcissistic personality disorder [5,6,7].

In early personality research, narcissism was defined using a single factor model. The current literature conceptualizes narcissism as a spectrum model consisting of the dimensions of grandiose and vulnerable narcissism anchored by a middle-ground dimension of self-entitlement/antagonism that both grandiose and vulnerable narcissism appear to share [3]. In addition, a tri-facet model has recently proposed the agentic (grandiose), antagonistic (self-entitlement), and neurotic (vulnerable) facets of narcissism, which mirror the spectrum in Krizan and Herlache’s [3] model [8,9]. Grandiose narcissism refers to the behaviors and feelings of exhibitionism, arrogance, and overconfidence [3,10,11]. Vulnerable narcissism refers to behaviors and feelings of defensiveness and hypersensitivity to criticism, withdrawal, avoidant behaviors, and sensitive attitudes [3,11,12,13,14]. While both dimensions of narcissism are characterized by self-absorption/self-importance/self-entitlement [3,8,9], with a need for the conservation of the ego, recent research has shown that these trait measures are distinct and have revealed distinctive features of both dimensions of narcissism in interpersonal and social situations. Examples of how the dimensions differ include self-esteem levels, threat perception, external and internal concerns, attachment style, altruistic/prosocial behavior, well-being and, within the Big 5-factor personality assessment, a specific focus on the factors of neuroticism, openness, and extraversion [3,11,15,16,17,18,19]. As the narcissism spectrum model (NSM) [3] continues to gain support, we anticipate further expansion of the conceptual model regarding linearity/non-linearity of the dimensions and determining characteristics and behaviors that distinguish the dimensions.

### 1.2. Vanity

Vanity originates from the Latin word vanus or vanitas, which describes something as “empty” or “meaningless” [20]. A similar term, vanite, found in the old French literature, translates into futility and self-conceit [20]. Both Latin and French contributed to our understanding of vanity. In English, the term embodies excessive self-concern and the idea that this concern or expectation may be subjective or meaningless to others beyond the self. As we use it, vanity is a concern for a positive view of one’s appearance and/or achievement; vanity has commonly been considered synonymous with excessive pride [21,22]. The conceptual and evolutionary foundations of vanity are set out in the social comparison theory developed by Festinger [23]: individuals determine their personal and social self-worth based on how they perceive themselves in contrast to others. People compare themselves to others within their social reference groups as they experience pressure toward uniformity within the reference group [23,24,25]. Individuals are motivated to make social comparisons to pursue self-evaluations for self-improvement and self-enhancement to boost self-esteem and self-concept [26,27]. Social comparison theory thus enables researchers to focus on the manifestations of self-concern, such as physical appearance and personal achievements. However, these are commonly subjective to an individual’s values and are volatile to external factors.

The first notable measure of vanity in personality appeared as a facet in the 40-item Narcissism Personality Inventory (NPI) [2]. However, this subscale contains only three items to assess vanity and only provides a limited evaluation of appearance vanity values [28]. Outside this sub-measure, there is a paucity of research on vanity in personality and behavioral psychology. Vanity is predominantly studied within consumerism, where social comparison theory applies to advertising methods and exposure [27,29]. Vanity is notably measured via two subtypes, appearance vanity and achievement vanity, which reflect the distinct types of personal goals of an individual along with perceptions of their appearance or achievements [22,30,31,32]. Specifically, we observe that an external predictor focuses on vanity and economic behavior and has limited consideration of the core motivations and underlying mechanisms of vanity in personality. A recent study suggested that individuals who self-report higher levels of vanity report higher levels of willingness to endure more external social costs and internal personal costs to achieve the public image they wish to attain. This highlights an important need to expand the personality literature to better understand the affective core of vanity and its patterns or relationships with social behavior in other personal areas of life [33].

### 1.3. Empathy, Narcissism, and Vanity

Hogan [34] described empathy as ‘‘the intellectual or imaginative apprehension of another’s condition or state of mind”, and, to the extent that it correlates with prosocial behavior, empathy is a core motivation of altruistic behavior [35,36,37]. Typically, narcissism prompts individuals to engage in public or communal acts of service to boost their self-valuation and affirm their grandiose perceptions about themselves, rather than stemming from empathetic values or altruism [38]. Narcissism has been characterized in research findings as self-centeredness and disregard toward others, which implies a lack of empathy or a deficit in individuals with high levels of narcissism [39]. Other factors may also co-occur (self-centeredness, self-esteem, self-control, motivation) that suppress empathic ability or result in a dysfunctional form of empathy that may register empathy as non-existent in the individual [39,40]. Higher levels of narcissism and impaired or empathy deficits may lead to relationship dysfunction, impaired social abilities, and an exaggerated self-concept within narcissistic individuals. However, the relationship between vanity (as opposed to narcissism) and empathy has not been well-studied. There is a gap in the existing literature regarding studies of vanity and the relationship that empathy may have with the more recent work positing the two types of vanity. This study examines the relationship between vanity and empathy.

### 1.4. Pride, Narcissism, and Vanity

Tracy and Robins [41] sought to broaden the understanding of pride in personality by outlining two types: authentic and hubristic. These are distinguished by the controllable, specific, or uncontrollable and global attributions that the type of pride reflects [42]. Authentic pride is based on something the individual did (controllable), bringing the individual a sense of accomplishment and pride. In contrast, hubristic pride arises from an individual’s concept of self (uncontrollable), derived simply from being and not based on specific actions [42]. Pride can be associated with both prosocial (authentic pride) and antisocial (hubristic pride) behaviors [43]. Hubristic pride and elevated levels of narcissism tend to lead to similar dysfunctional social outcomes, as each is positively associated with adverse relational outcomes, antisocial behaviors and perceptions of others, and aggression [44,45]. Alternatively, authentic pride and genuine self-esteem have positive associations with successful relationships, prosocial outcomes and perceptions, and positive mental health outcomes [44]. Unfortunately, there is a paucity of research on vanity and its associations with pride, as the literature has restricted each to their implied synonymy. Our study seeks to fill this gap in the literature regarding vanity (appearance and achievement) and its relationship with pride (authentic and hubristic).

### 1.5. Social Behaviors of Interest Regarding Narcissism and Vanity: Sensitivity to Others, Communal Image Management, and Selflessness

“Sensitivity to Others” is a concept rooted in sociality and tied to self-autonomy [45,46]. This social concept mirrors how reactive or “sensitive” one is to the thoughts, feelings, intimacy, and needs of others [45,46]. Sensitivity to others consists of cognitive and behavioral processes about how quick one is to react in providing a need or support to another person and, thus, sustaining their own personal and social need for relationships. Conceptually, sensitivity to others provides the awareness and inclination under which social connectedness and bondedness may occur [47]. Accordingly, sensitivity to others was assessed as a criterion variable to determine whether vanity and narcissism differed in self-reports of social sensitivity and reactivity.

Communal image management is the management of one’s self-presentation to reflect community morale and pleasantry [48]. It consists of qualities of self-presentation that help with “getting along” within a community to which one belongs [48,49]. We chose communal image management to determine whether vanity and narcissism differ in their associations with one’s efforts at self-presentation or promote communal morale for the sake of one’s identity and, importantly, whether based on an actual audience or an “internalized” audience [50].

Selflessness is the ability to push aside needs and interests to fulfill or serve the interests of others [51]. Selflessness has been studied in research focused on social likeability, relational satisfaction, and so on [49]. We chose selflessness to determine whether vanity and narcissism differed in their associations with the inclination or ability to be selfless.

### 1.6. Purpose, Study Outline, and Hypotheses

Narcissism and vanity are presumed to resemble one another in terms of connotations and usage in communication, and both incorporate self-centered qualities and concerns. Narcissism (grandiose and vulnerable) is also characterized by a lack of concern for others and a sense of entitlement [2,3,52], whereas vanity is less well specified in comparison in personality literature, although it has been characterized as a drive to increase or highlight one’s value of self with an additional emphasis on concern about one’s public image [21,33]. The single-factor narcissism model traditionally conceptualizes having ‘vanity’ as a facet of narcissism [2]. This is observed in the widely used Narcissism Personality Inventory [2], which measures vanity using a three-item assessment. The recent reevaluation of narcissism as a dimensional or spectrum model [3,8,9] prompted us to reconsider the validity of vanity as a facet of the newer conceptual model of narcissism, as trait measures of grandiose and vulnerable narcissism are distinct [12,13,53]. A recent study found that while single-factor narcissism revealed a strong relationship with the NPI vanity facet, grandiose narcissism revealed a small relationship with NPI vanity and vulnerable narcissism revealed a non-significant relationship with NPI vanity [28]. In addition, a more comprehensive vanity scale [22] has been established since the introduction of the NPI and warrants the examination of narcissism using a potentially better measure of vanity. Modern developments in both the literature and refinement of methodologies that examine both vanity and narcissism challenge the traditional and more outdated understanding of how the two interrelate. This raises the question we want to explore in this paper: would a more comprehensive measure of vanity reveal associations with both or one of the dimensions of narcissism and therefore remain a facet of the new narcissism model, or may it be the case that vanity is distinct from the dimensions of narcissism as outlined by more recent work? As the topic of narcissism in the psychological community has increased in popularity and contributes value to understanding the self, this calls for a need to further expand the narcissism model and its relationship to other personality constructs and behaviors [8]. It makes sense to reexamine vanity’s relationship to narcissism as we reassess what is contributing to the characteristic foundation of the dimensions of narcissism. Therefore, we seek to challenge and examine how vanity and narcissism overlap or may not overlap based on current conceptions of the two. Unraveling a potential new relationship between vanity and narcissism will reveal implications related to how we understand the “motivational dynamics” of both narcissism and vanity [8]. Given that these core motivations influence social behavior, we suspect that reevaluating the recent conceptualizations of vanity and narcissism will reveal novel implications associated with the traits and how they influence social behavior.

We examined how narcissism and vanity may interact as a function of how each construct relates to other social characteristics to determine whether vanity and narcissism are differentiated. For the selection of the social behaviors to study, we selected social outcomes that are commonly associated as being influenced by narcissism or have been commonly studied relative to personality and the concern with self. These include pride (authentic and hubristic), empathy, sensitivity to others, image management, and selflessness. Our study aims to determine where vanity may reside in personality relative to narcissism and expand on vanity in personality psychology, where a gap in the literature is present.

We hypothesized (H1) that narcissism and vanity are distinguishable and independent multidimensional constructs. We hypothesized (H2) that self-reported narcissism and vanity will differ in their associations with social characteristics, such as pride and empathy, and by way of the selected social outcome variables. We hypothesized (H3) that both types of self-reported vanity (appearance and achievement) will have a negative relationship with empathy and correlate with higher levels of self-reported authentic pride. We hypothesized (H4) that both types of self-reported narcissism will lack a relationship with self-reported empathy and higher levels of self-reported hubristic pride than vanity. However, we also predict that self-reported grandiose narcissism will evidence higher levels of authentic and hubristic pride based on previous literature on single-factor narcissism and pride [33,42,44,51,54]. Based on the lack of preexisting work on vanity and personality, our study concerning social outcome variables is exploratory. However, we hypothesized (H5) that the dimensions of narcissism will show significant relationship patterns with sensitivity to others, selflessness, and communal image management.

## 2. Materials and Methods

### 2.1. Participants

Participants were 441 undergraduate students from a large public midwestern university enrolled in PSY101 coursework. The sample consisted of 329 females, 102 males, and 10 participants who did not disclose their gender or self-reported as non-binary. There were no differences by gender on any of the primary predictor variables, and, as a result, we did not perform any analyses by gender. The mean age was 19.0 (SD = 2.08). The racial/ethnic composition of our sample was reported to be 78.3% white, 8.9% Hispanic or Latino, 5.3% African American or Black, 3.4% Asian or Pacific Islander, 3.1% multi-ethnic, less than 1% reported as American Indian or Alaskan Native, and less than 1% did not report their racial/ethnic identity.

### 2.2. Measures/Instruments

The Narcissism Personality Inventory-16 [2,55] or NPI-16 assesses subclinical narcissism, considering the traditional characteristics of narcissism associated with grandiose narcissism (exhibitionism, grandiosity, self-entitlement, etc.). The NPI-16 consists of 16 paired items, with 0 scoring if a participant chooses the neutral statement and 1 scoring if the grandiose statement is chosen. An example of a grandiose item from this measure states, “I like to be the center of attention”, with its counterpart as “I prefer to blend in with the crowd”. After reverse scoring the requisite items, NPI-16 scores were obtained by summing the total survey responses of the participants; the score represents the number of grandiose narcissism characteristics self-reported as present. Cronbach’s alpha for the Narcissism Personality Inventory was reported as 0.85 [55].

The hypersensitive narcissism scale (HSNS) [56] assesses subclinical narcissism characteristics associated with vulnerable narcissism (e.g., defensiveness and self-absorption). The HSNS consists of 10 5-point Likert scale items, with 1 scoring as “strongly disagree” and 5 scoring as “strongly agree”. An example of an item from this measure states, “I dislike being in a group unless I know that I am appreciated by at least one of those present”. Scores were obtained by summing the participants’ survey responses; the score represented the number of vulnerable narcissism characteristics self-reported as present. Cronbach’s alpha for the hypersensitive narcissism scale has been reported as 0.72 and 0.76 for females and males, respectively, scoring the male data sample from Cheek and Melchior’s (1985) data (as cited [56]).

The vanity scale [22] consists of two subscales of vanity (appearance and achievement) to assess the trait across four total dimensions: appearance concern, appearance view, achievement concern, and achievement view. The vanity scale consists of 21 7-point Likert-scale items, with 1 scoring as “strongly disagree” and 7 scoring as “strongly agree”. An example of an appearance vanity item from this measure states, “The way I look is extremely important to me”. An example of an achievement vanity item from this measure states, “I am more concerned with professional/academic success than most people I know”. Scores were obtained by summing the total of the participants’ survey responses to the subscales of appearance vanity and achievement vanity, and the scores of each subscale represented the amount of appearance and/or achievement vanity characteristics self-reported as present. Cronbach’s alpha for the vanity scale has been reported across the four scale dimensions ranging from 0.80 to 0.92 [22].

The pride short-form scale [42] assesses how closely one may generally feel daily towards descriptive words that describe authentic and/or hubristic pride based on self-reporting. The scale consists of 14 items rated on a 5-point Likert scale ranging from 1 (not at all) to 5 (extremely). An example of an authentic pride item from this measure states, “Rate to the extent you generally feel accomplished”, and an example of a hubristic pride item states, “Rate to the extent you generally feel smug”. Cronbach’s alpha for the pride short-form scale has been reported as 0.88 for the authentic pride scale and 0.90 for the hubristic pride scale.

The Jackson Personality Inventory: Empathy [57], or JPI: EMP assessed self-reported levels of empathetic ableness toward others. The full-length JPI consists of 300 true/false items, with 15 subscales of personality domains. We used items from the empathy subscale. An example of an item from the empathy subscale states, “I feel others’ emotions”. Cronbach’s alpha for the Jackson Personality Inventory: Empathy has been reported as 0.80 [57].

The autonomy–connectedness scale [46], or ACS, assesses “self-awareness, sensitivity to others, and the capacity for managing new situations”. The ACS consists of 30 items with three subscales to measure the respective attributes. The items were measured on a 5-point Likert scale ranging from 1 (disagree) to 5 (agree). We used items from the subscale of “sensitivity to others”. An example of an item from the sensitivity to others subscale states, “I can hardly bear it when people are angry with me”. Cronbach’s alpha for the autonomy–connectedness scale has been reported as 0.78 for the subscale of “sensitivity to others” [46].

The bidimensional image/impression management scale [58], or IM, assesses the distinction between agency and communal image management of one’s self-presentation based on self-reports. The IM short form consists of 20 items derived from the SDR [59] and forms the subscales of agentic management (AM) and communal management (CM) [48]. The items were measured on a 7-point Likert scale ranging from 1 (not true) to 7 (very true). We used the communal image management subscale to assess image management within a social/communal context. An example from the communal image management subscale states, “I have done things that I don’t tell other people about”. Cronbach’s alpha for the bidimensional image management scale, specifically the communal management subscale, ranged from 0.75 to 0.91 [60].

The selflessness scale (SS) [3] assesses the perception of one’s ability to act selflessly based on self-reports. The SS consists of 15 items rated on a 4-point Likert scale ranging from 1 (highly disagree) to 4 (highly agree). An example of an item from the scale states, “I am willing to sacrifice a lot for the benefit of others”. Cronbach’s alpha for the selflessness scale was 0.66 [49].

### 2.3. Procedure

Students from undergraduate psychology courses at the university were asked to participate in an online study using a Qualtrics survey via SONA Systems. The survey included 116 multiple-choice questions, and each scale was scored on a 5-point Likert scale, scoring (1) strongly disagree to strongly agree (5), a 7-point Likert scale scoring (1) strongly disagree to strongly agree (7), or as a true/false scale with (0) false and (1) indicating that the statement was true. We estimated that participants would require approximately 20–30 min to complete the self-report-based survey. The IRB required that participant background information be obtained before opening the survey portion and informed consent was obtained before the initiation of the survey. Upon completing the survey, each participant was debriefed, thanked, and received credit for their designated psychology course. Results from the questionnaires were then organized and analyzed using SAS software, Version 9.4 of the SAS System for Unix [61]. Copyright © 2020 SAS Institute Inc. SAS and all other SAS Institute Inc. product or service names are registered trademarks or trademarks of SAS Institute Inc., Cary, NC, USA.

### 2.4. Data Analysis

The results from the questionnaires were organized and analyzed using SAS software, Version 9.4 of the SAS System for Unix [61]. Correlation analysis was performed for each respective independent variable with every dependent variable to assess the potential relationships between our traits of interest and the social characteristics selected. Based on our study design, we checked for the possibility of common method variance (CMV). Per Eichhorn (pp. 6–7) [62], we used SAS to run a non-rotated principal single-factor test to perform Harman’s single-factor test. The Harman single-factor technique estimated CMV to be 25.2%. This did not exceed the commonly accepted threshold of 50% and suggested that CMV was not influencing our findings. Regression analysis was performed to control for shared variance in our predictor variables based on Tracy et al., (p. 208) [44] and should be interpreted in such a manner.

## 3. Results

All the correlations, means, and alphas (values located in parentheses in each respective variable column represents the alpha value) are listed in Table 1. Both self-reported appearance vanity and achievement vanity revealed significant positive relationships with grandiose narcissism. In addition, self-reported appearance vanity and achievement vanity were positively associated with vulnerable narcissism. In accordance with our predictions, self-reported vulnerable narcissism and grandiose narcissism revealed positive relationships with hubristic pride. Our results also confirmed our prediction that an individual with vulnerable narcissism would report a significant negative relationship with authentic pride. In contrast, self-reported grandiose narcissism showed a significant positive relationship with authentic pride.

As predicted, both subtypes of vanity were associated with significant levels of authentic pride; participants who self-reported achievement vanity and appearance vanity revealed positive relationships with authentic pride. Differences in the correlations were reported as significant, with self-reported appearance vanity evidencing a stronger relationship with authentic pride (z-score = 2.82, *p* ≤ 0.01). Self-reported achievement vanity also had a small positive association with hubristic pride. Differences between the correlations of achievement vanity and hubristic pride and vulnerable narcissism and hubristic pride were non-significant. Self-reported appearance vanity was not significantly associated with hubristic pride.

As noted, prior work has found that people reporting higher levels of trait narcissism lack the ability to empathize or endorse empathetic values. Our prediction aligns with the finding of a slight inverse relationship between self-reported grandiose narcissism and empathy. However, the finding of a slight positive correlation between self-reported vulnerable narcissism and empathy contradicts our predictions. We predicted that people’s self-reported vanity would also report empathy. However, the results revealed that self-reported achievement and appearance vanity had no significant relationships with empathy.

We then considered the predictors of narcissism and vanity with theoretical social behavior outcome variables. Individuals who reported grandiose narcissism were found to have a significant negative relationship with sensitivity to others. In contrast, self-reported vulnerable narcissism positively correlated with sensitivity to others. Individuals with self-reported appearance vanity and achievement vanity had non-significant relationships with sensitivity to others. Self-reported grandiose narcissism and vulnerable narcissism revealed significant negative relationships with communal image management). Likewise, self-reported achievement vanity and communal image management revealed a slightly negative relationship, whereas self-reported appearance vanity was not significant. While individuals with vulnerable narcissism reported a small negative relationship with selflessness, self-reported grandiose narcissism revealed a much stronger negative association with selflessness. Participants who reported higher levels of achievement vanity and appearance vanity had non-significant relationships with selflessness.

A series of regressions were run on pride and empathy, where we simultaneously entered grandiose narcissism, vulnerable narcissism, appearance vanity, and achievement vanity as predictors, as shown in Table 2. This aimed to control for any potential shared variance that these predictors might have had in the correlation analysis. For empathy, self-reported grandiose narcissism as a negative predictor held in the regression, whereas the relationship between self-reported vulnerable narcissism and empathy was reduced to non-significance. Self-reported achievement and appearance vanity remained non-significant to empathy. For authentic pride, self-reported vulnerable narcissism held as a negative predictor in the regression, and self-reported appearance vanity and grandiose narcissism held as positive predictors. Achievement vanity’s relationship to authentic pride was reduced to non-significance. Regarding hubristic pride, self-reported grandiose and vulnerable narcissism remained positive predictors, while achievement vanity was reduced to non-significance. Appearance vanity remained non-significant.

Another series of regressions were run on all social behavior outcome variables. We simultaneously entered grandiose narcissism, vulnerable narcissism, appearance vanity, and achievement vanity as predictors, as shown in Table 3. This was executed to control for any potential shared variance between these predictors, as discussed in the previous regression. Regarding sensitivity to others, self-reported grandiose narcissism remained a negative predictor, whereas self-reported vulnerable narcissism remained a positive predictor. Self-reported achievement vanity remained non-significant for sensitivity to others. Interestingly, self-reported appearance vanity became a positive predictor for sensitivity to others. For selflessness, self-reported grandiose and vulnerable narcissism remained negative predictors, while vanity predictors remained non-significant. Regardingcommunal image management, self-reported grandiose and vulnerable narcissism remained as negative predictors. However, the relationship between achievement vanity and communal image management was reduced to non-significance, while appearance vanity remained non-significant in the regression model.

## 4. Discussion

We discuss four main points related to our findings that pertain to our research question: (1) Vanity has a distinct characterization but empirically lacks the broadness and depth that narcissism has as a personality construct; (2) Our findings reaffirm the current conception of narcissism; (3) Vanity’s correlations with both dimensions of narcissism appear to support retaining vanity as a facet of narcissism, and we determine where vanity may fall as a facet concerning the spectrum model of narcissism; (4) Lastly, we discuss the theoretical implications of the study that reflect the motivational cores of vanity and narcissism.

### 4.1. Characterization of Vanity: Pride and an Absence of Social Behavior

Authentic pride was the main characteristic we identified as significantly associated with self-reported vanity. When a participant self-reported elevated levels of vanity, those participants also self-reported higher levels of “confidence”, “self-worth”, “fulfillment”, etc., from the subscale of authentic pride [42]. Goals or ideals established based on vanity may allow individuals to celebrate actions that bring them pride when accomplished. The regression confirmed in our data that appearance vanity is independently associated with authentic pride, while achievement vanity was reduced to non-significance. We consider that a sense of authentic pride in individuals may be more associated with appearance vanity than achievement vanity. On the downside, achievement vanity and hubristic pride also evidenced a slight positive correlation, though this was reduced to non-significance in the regression. We speculate that this small correlation reveals a shared variance between self-reported achievement vanity and a predictor of narcissism, specifically vulnerable narcissism, as vulnerable narcissism resulted as a positive predictor of hubristic pride. Overall, this finding supports the positive relationship between vanity and authentic pride, though we suggest that further research be conducted regarding distinguishing the vanity subtypes and pride types. This is consistent with previous observations that vanity and pride share similar connotations [21,63]. While we would argue against vanity and pride being synonymous based on their differing underlying motivations, the fact that pride is classified as an emotion supports authentic pride as internalized via vanity. Vanity is driven by a core motivation of intrinsic goals to enhance one’s appearance and/or achievement status. If one perceives those goals to be achieved, a result of authentic pride will occur. 

In addition to a characterization of pride, we found both vanity subtypes to be characterized by a lack of empathy and a lack of reported relationships with the selected social behavior variables. While self-reported achievement vanity revealed a slight correlation with communal image management, this was reduced to non-significance in the regression and is suggestive that shared variance from other predictors resulted in the previous correlation. Interestingly, a non-significant correlation between self-reported appearance vanity and sensitivity to others became a positive predictor–outcome relationship when the regression was run; this suggests that the appearance vanity subtype may be characterized by sensitivity to the reactions and opinions of others where achievement vanity is not and became evident once shared variance between predictors was controlled for. This finding or, rather, the lack of overall findings with social outcomes, suggests that the construct of vanity appears to lack the empirical support to justify considering it as a distinct personality construct. 

### 4.2. Revisiting the Characterization of Narcissism

When assessing the dimensions of narcissism, our results largely corroborate previous literature. Our hypotheses were correct concerning the facets of pride (see H3 of Section 1.6). While grandiose narcissism reported an expected inverse relationship with empathy, vulnerable narcissism revealed a positive relationship with empathy. Interestingly, the association between self-reported vulnerable narcissism and empathy is consistent with previous findings; vulnerable narcissists tend to self-report empathy [64]. Previous literature finds that the type of empathy a vulnerable narcissist can exhibit is often associated with maladaptive aspects of empathy, such as maladaptive affective empathy or empathic distress [64,65]. Our measure included items with a central theme of internalizing others’ distressing situations in empathic ability [57]. Items such as “suffer from others’ sorrows” or “easily moved to tears” suggest the propensity to experience emotional distress based on someone else’s experience. Thus, in our study, vulnerable narcissists with self-reported empathy may be constrained by the dysfunctional reality of their limited abilities as they pursue their need for connections, albeit with motivation for external validation from others [18,64]. 

Concerning the social outcome variables we selected, the results reaffirmed our understanding of the dimensions of narcissism and how they may differ or be alike. While grandiose narcissism revealed a self-reported inverse relationship with sensitivity to others, self-reported vulnerable narcissism revealed a positive relationship. Individuals with vulnerable narcissism are characterized as defensive and seek external validation due to low self-esteem and a high likelihood of insecure attachment [8,11]. This is consistent with the idea that an individual with vulnerable narcissism would have a heightened sensitivity to others to seek the affirmation they need to affirm their insecure sense of self [3,12,13], whereas the depiction of grandiose narcissism is the opposite, with a high self-esteem reported and a sense of security within oneself [3,8,12]. Regarding selflessness, the inverse relationships for both dimensions of narcissism leads us to conclude that these reflect the self-entitlement that both grandiose and vulnerable share as the middle ground of the narcissism spectrum model, where self-entitlement is characterized by arrogance and self-centeredness [3]. The dimensions of narcissism also evidence communal image management, where their inverse relationships with communal image management suggest that they are avoidant to admitting wrongdoing or immoral behavior. Overall, our findings are aligned with the current conception of narcissism as a spectrum model.

### 4.3. Vanity’s Relationship to Narcissism: Where Does It Fall on the Narcissism Spectrum Model?

While vanity can be identified with characterizations of authentic pride, we believe that the lack of relationship patterns with the various social characteristics we assessed is evidence of a lack of depth to vanity in the realm of personality and does not appear to be a distinct construct apart from narcissism in personality. Personality constructs, such as narcissism, are stable and succinct predictors of other social characteristics we possess, where in turn, personality heavily influences our social behavior. Our results are in line with previous findings such that vanity, in personality, would be classified as a facet of narcissism, and this remains true using the most recent conceptual model of narcissism [3]. This is supported by our finding that both subtypes of vanity revealed significant positive relationships with both dimensions of narcissism while lacking relationships with most social outcome variables considered.

The concluding aim of our discussion is to consider where vanity best aligns with the narcissism spectrum model. Based on the results, we propose that both subtypes of vanity align best with grandiose narcissism. Self-reported grandiose narcissism and both subtypes of vanity reveal positive relationships with authentic pride and positive relationships with hubristic pride for grandiose narcissism and achievement vanity only. While self-reported vulnerable narcissism also has a relationship with hubristic pride, we find support that vanity and grandiose narcissism share more similarities after considering their relationships with *both* facets of pride.

Vanity as a facet of grandiose narcissism reflects a potentially improved theoretical understanding of the underlying mechanism behind grandiose narcissism’s relationship with pride and how this contributes to its traditional characterization. The core motivation of vanity, based on social comparison theory, helps us best explain this possible mechanism. Because an individual with grandiose narcissism is identified with extraversion and exhibitionism, it would make sense to utilize social comparison (vanity) to monitor their grandiose image, self-esteem, and self-concept. Suppose grandiose individuals perceive their appearance or achievement as satisfactory based on their self-evaluation relevant to their reference group. In that case, they experience authentic pride and feel accomplished via their actions and behaviors. Their grandiosity and vain concerns with themselves fuel hubris, as evidenced in our results. While they may feel accomplished based on a specific goal that was met regarding their appearance or achievement, and evidence of authentic pride, this will also reaffirm their grandiose perceptions of themselves and the potential to develop hubris. Thus, a cycle arises: vanity utilizes social comparison to present an opportunity for a sense of authentic pride, which then can feed a grandiose perception of self that then reaffirms a hubristic and global pride about oneself, complementing their extraversion and high self-esteem. This characterization driven by vanity is best suited to grandiose narcissism, as an individual with vulnerable narcissism evidences hubristic pride because they are guided by shame and defensiveness of themselves and lack a sense of authentic pride. These theoretical implications further strengthen our view of vanity as a facet of grandiose narcissism.

The proposal of vanity as a facet of grandiose narcissism is supported by the previous literature as well as established work. The study by Brown [28] that we base our research question on revealed in their findings that the NPI sub-measure of vanity evidenced a stronger correlation with their grandiose narcissism measure than their measure for vulnerable narcissism. Utilizing a more comprehensive measure of vanity, we provide further support for vanity as a facet of narcissism aligned with the grandiose dimension of the narcissism spectrum model.

### 4.4. Theoretical Implications Regarding the Motivational Dynamics of Vanity and Narcissism 

The lack of a relationship between the selected social outcomes and vanity fosters a deeper understanding of the motivational core of vanity. It may be that vanity’s deep focus on intrinsic personal goals and values maintains a desired self-image (whether in appearance or one’s achievement status) and contributes to the social behavior void observed; this suggests that vanity is primarily motivated by the individuals themselves to meet their own needs and goals, in contrast to narcissism that is historically socially motivated to meet the individual’s needs and goals. 

Social comparison theory, as discussed in the introduction, is based on the assumption that individuals determine their self-worth based on how they perceive themselves relative to others in close peer groups and is indicative of vanity [23,24,25]. While they may be utilizing social comparison to assess their own status relative to their appearance and/or achievement goals, the act itself is not objectively social. Rather, the individual is motivated on their own account of what they perceive as a need or a want for their own ideals and uses their social environment as a baseline to assess where they stand with their expectations and desires. 

Narcissism, on the other hand, reflects a social motivation at its core in our finding of identified relationships with the selected social outcomes. Indeed, individuals with elevated levels of narcissism exhibit social activity as narcissism itself seeks social validation to maintain itself. While narcissism is a focus on the self, sometimes at an exaggerated and maladaptive level, it is rooted in being motivated socially to affirm one’s preferred feelings about oneself. Simply put, both dimensions of narcissism are part of a socially contingent personality construct that motivates individuals to manage their self-image and self-concept via social experiences and interactions. While it is not inaccurate to characterize both vanity and narcissism with self-involvement, it is important to discuss how these differ at their core. We conclude that vanity is best retained as a facet of grandiose narcissism, especially within the narcissism spectrum model. One key contribution of our study is the improved understanding of the motivational dynamics of vanity and narcissism, in particular our understanding of how the two interact and, thus, contribute to personality.

### 4.5. Limitations and Future Research

A few aspects limit the research. With evidence supporting vanity to be stronger linked to grandiose narcissism, we do acknowledge that in our study we did not include a measure for self-entitlement. Self-entitlement is the middle ground point of the narcissism spectrum model in addition to the grandiose and vulnerable dimensions of narcissism [3]. It is becoming more widely measured in addition to grandiose and vulnerable narcissism scales. While we sought to examine how vanity would relate to both profoundly distinct dimensions of narcissism, we recognize that examining the relationship between self-entitlement and vanity would be beneficial. We encourage future research on self-entitlement and vanity to understand further how vanity aligns with the narcissism spectrum model. We acknowledge that in a self-report design survey regarding personality constructs, the participants’ responses are subject to self-enhancement and halo effect biases [66]. The age of the participant group is a potential limitation; younger individuals are more likely to be influenced by external opinions as well. Our participants are all young undergraduate students who may face additional environmental influences. Thus, results may have revealed different findings if the study had been performed in a different sample. Future research would benefit from considering alternative data collection methods to assess narcissism and vanity. Future research should also consider vanity, other types or traits of social behavior, and social scenarios to further contribute to vanity’s characterization (or lack thereof) of social behavior.

## 5. Conclusions

Our data support the retention of vanity as a facet of narcissism rather than as a distinct personality construct based on its lack of breadth and depth relative to social characteristics. Furthermore, we found that vanity was associated with grandiose narcissism, which has been anticipated and confirmed by previous investigations. Overall, our integration of newer comprehensive models of vanity and narcissism clarifies how vanity fits into the field of personality. Our clarification of vanity is necessary to understand vanity in personality and its core motivations and to expand and clarify the narcissism spectrum model. The findings add to the popular topic of personality psychology, ultimately enhancing our understanding of narcissism, vanity, and its translation to social behavior.

## Figures and Tables

**Table 1 behavsci-13-00762-t001:** Means, Cronbach alphas, and correlations between narcissism, vanity, pride, empathy, bondedness, selflessness, and communal image management.

Criterion	Gran. Narc.	Vuln. Narc.	Ach. Vanity	App. Vanity	Empathy	A. Pride	H. Pride	Sensitive	Selfless	IM
Grandiose narcissism	[0.65]									
Vulnerable narcissism	0.02	[0.64]								
Achievement vanity	0.34 ***	0.21 **	[0.85]							
Appearance vanity	0.26 ***	0.18 **	0.38 ***	[0.85]						
Empathy	−0.14 *	0.13 *	0.06	0.05	[0.76]					
Authentic pride	0.35 ***	−0.21 **	0.24 ***	0.41 ***	−0.06	[0.86]				
Hubristic pride	0.34 ***	0.19 **	0.14 *	0.08	−0.17 **	0.16 *	[0.87]			
Sensitivity to others	−0.25 ***	0.18 **	−0.06	0.04	0.55 ***	−0.24 ***	−0.22 **	[0.83]		
Selflessness	−0.24 ***	−0.13 *	−0.05	−0.03	0.38 ***	−0.07	−0.22 **	0.49 ***	[0.69]	
Communal image management	−0.21 **	−0.27 ***	−0.14 *	−0.09	0.004	0.12 *	−0.21 **	−0.10*	0.20 **	[0.67]
Means	20.4	30.8	46.1	44.3	16.9	20.2	12.2	55.7	36.2	33.4
SD	2.8	5.2	10.6	9.2	2.5	4.7	4.7	8.7	4.9	8.6

Note: All scales have N = 441. Gran. Narc = Grandiose narcissism; Vuln. Narc. = Vulnerable narcissism; Ach. Vanity = Achievement vanity; App. Vanity = Appearance vanity; A. Pride = Authentic pride; H. Pride = Hubristic pride; Sensitive = Sensitivity to others; Selfless = Selflessness; Communal IM = Communal image management. * *p* < 0.05 ** *p* < 0.001 *** *p* < 0.0001.

**Table 2 behavsci-13-00762-t002:** Regression analyses controlling for shared variance and predicting pride and empathy traits related to narcissism and vanity.

Dependent Variable	Grandiose Narcissism	Vulnerable Narcissism	Appearance Vanity	Achievement Vanity
Empathy	−0.15 ** (0.04)	0.06 (0.02)	0.01 (0.01)	0.02 (0.01)
Authentic pride	0.38 ** (0.07)	−0.27 ** (0.04)	0.19 ** (0.02)	0.03 (0.02)
Hubristic pride	0.56 ** (0.08)	0.18 ** (0.04)	−0.02 (0.02)	0.001 (0.02)

Note: All scales have N = 441. ** *p* < 0.001.

**Table 3 behavsci-13-00762-t003:** Regression analyses controlling for shared variance and predicting theoretical constructs related to narcissism and vanity.

Dependent Variable	Grandiose Narcissism	Vulnerable Narcissism	Appearance Vanity	Achievement Vanity
Sensitivity to others	−0.30 * (0.13)	0.16 * (0.05)	0.12 * (0.03)	−0.05 (0.03)
Selflessness	−0.27 ** (0.08)	−0.17 ** (0.05)	0.02 (0.02)	0.009 (0.02)
Communal image management	−0.60 ** (0.15)	−0.29 ** (0.08)	−0.06 (0.03)	−0.04 (0.03)

Note: All scales have N = 441. * *p* < 0.05 ** *p* < 0.001.

## Data Availability

All data are available upon request.

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
