# Peer review of "A Narcissism/Vanity Distinction? Reassessing Vanity Using a Modern Model of Narcissism Based on Pride, Empathy, and Social Behaviors"

_behavsci, 2023, doi:10.3390/bs13090762_

Round 1
Reviewer 1 Report
In this exploratory study, the authors seek to assess the overlap between vanity and narcissism. The study is well-written and has all the necessary information. However, I have a few comments for the authors to consider:
- I suggest the authors use keywords that do not appear on the title, thus ensuring a broader manuscript reach once published.
- Please provide participants’ descriptions in the abstract. I also suggest that the authors further improve the information presented in the abstract to more concrete findings instead of using broader sentences.
- I highly encourage the authors to include a data analysis section explaining which analyses were performed and based on which reference they should be interpreted.
- Because the authors used a particular sample (i.e., undergraduate students), I suggest they include in their limitations how narcissism and vanity can have an impact, considering participants’ age. Young adults can be highly affected by external opinions. Thus, results could have been different if the study was performed in a different sample.
Reviewer 2 Report
Thank you very much for the opportunity to review an interesting article on narcissism. I have just a few comments for the authors:
1. INTRODUCTION
- Perhaps it would make sense to merge "1.1 Purpose", "1.4 Research question, study outline, and aims" and "1.10 Research Hypotheses" and move it to the end of the chapter?
- Some of the subchapters are very short and somewhat interrupt the flow of reading. Could you consider merging them?
2. MATERIALS AND METHODS
- Could the order of the Procedure and Masures/Instruments chapters be reversed?
3. RESULTS
- 2nd line (and everywhere else): please write p accordingly (p < .0001), without the equaliser
Reviewer 3 Report
Review of the manuscript, entitled “A narcissism/vanity distinction? Reassessing vanity using a modern model of narcissism based on pride, empathy, and so[1]cial behaviors”.
Manuscript ID: behavsci-2580837
Critical Review
This is a cross-sectional study in the USA sample of 441 undergraduate students in psychology, consisting of 441 participants (321 women). The mean age was 19.0 (SD = 2.08). The aim of the study was “to determine where vanity may exist in personality relative to narcissism and expand on vanity in personality psychology”. The authors asked whether “vanity remain better construed as a facet of narcissism, or is it now better conceptualized as a distinct construct separated from earlier models of narcissism”. Based on traits of pride, empathy, and several social behavior variables several hypotheses were stated. Correlation and regression analyses were conducted. A differentiation between narcissism and vanity was expected, and found. Results revealed that vanity was limited to pride with an absence of social behavior. It is important because vanity is not socially aversive. The authors discuss an improved understanding of the narcissism model, and concluded that vanity is more closely related to the grandiose dimension of narcissism and not to the vulnerable narcissism.
I will now try to answer the twin question: “Is this study new and is it true?”
Is this study new? I have made a search in existing data bases. The results of this searching indicates that this issue is novel. By reading the paper, it’s clear why there is a need for the study being reported. The results are presented in a clear and simple way, and I may agree with the authors’ conclusions. I really like this study. It is well-argued, well-written, and the scales are well-described.
The next question is if this study is true, that is, the question concerns the methods used. I think that more work should be given for the presentation of the methods and results (see my suggestions and recommendations below). In general, before publication, the Result section should be improved.
I have several comments, recommendations and suggestions, without ranking them based on their importance.
#1 The study is on young people studying psychology. This is an important limitation, which should be discussed.
#2 Procedure. The reader may be confused because some scales are scored 1 = strongly disagree, while others are 1 = is true. This may be difficult to keep this in mind and to read, and understand the results (see tables). I think that it should be easier for the reader if the authors could transform the scales in the same direction.
#3 With so many scales, correlated with each other, the authors should check for, and describe the result of the Common Method Variance.
4# Table 1. The authors should explain in the note what do the figures in the parentheses mean (I understand, but it should be explained).
5# P. 8. The numerical values in the text should be deleted. The reader may see these results in the table.
6#. Tables 2 and 3. The results from the regression analyses should be presented in an easier way. Reader does not know what the figures in their tables represent. The way the authors chosen for presenting their results are unconventional. For example, Andy Field in his statistical book “Discovering statistics using IBM SPSS statistics”, 8.9 p. 352, gives an example how to report in tables results from regression analyses. The authors should report B, SE B, beta, p, and also R2 etc.
7# Table 3. The authors should check their table note. Why they did explain A and H? This must be from an early version.
Thank you for the opportunity to read this interesting manuscript. I hope that my comments, recommendations and suggestions may improve the quality of this manuscript.
